

# Identification of genes and functional coexpression modules closely related to ulcerative colitis by gene datasets analysis

Jie Zhu[1], Zheng Wang[1], Fengzhe Chen[1] and Changhong Liu[2]

[1] Department of Infectious Diseases, Qilu Hospital, Shandong University, Jinan, Shandong, China
[2] Department of Gastroenterology, Shandong Provincial Qianfoshan Hospital, the First Hospital Affiliated with Shandong First Medical University, Jinan, Shandong, China

## ABSTRACT

**Background**. Ulcerative colitis is a type of inflammatory bowel disease posing a great threat to the public health worldwide. Previously, gene expression studies of mucosal colonic biopsies have provided some insight into the pathophysiological mechanisms in ulcerative colitis; however, the exact pathogenesis is unclear. The purpose of this study is to identify the most related genes and pathways of UC by bioinformatics, so as to reveal the core of the pathogenesis.

**Methods**. Genome-wide gene expression datasets involving ulcerative colitis patients were collected from gene expression omnibus database. To identify most close genes, an integrated analysis of gene expression signature was performed by employing robust rank aggregation method. We used weighted gene co-expression network analysis to explore the functional modules involved in ulcerative colitis pathogenesis. Besides, biological process and pathways analysis of co-expression modules were figured out by gene ontology enrichment analysis using Metascape.

**Results**. A total of 328 ulcerative colitis patients and 138 healthy controls were from 14 datasets. The 150 most significant differentially expressed genes are likely to include causative genes of disease, and further studies are needed to demonstrate this. Seven main functional modules were identified, which pathway enrichment analysis indicated were associated with many biological processes. Pathways such as 'extracellular matrix, immune inflammatory response, cell cycle, material metabolism' are consistent with the core mechanism of ulcerative colitis. However, 'defense response to virus' and 'herpes simplex infection' suggest that viral infection is one of the aetiological agents. Besides, 'Signaling by Receptor Tyrosine Kinases' and 'pathway in cancer' provide new clues for the study of the risk and process of ulcerative colitis cancerization.

# INTRODUCTION

Ulcerative colitis (UC) is a subtype of inflammatory bowel disease (IBD), which is a kind of idiopathic, chronic, recurrent, debilitating and nonspecific inflammatory condition, and its characteristic is the alternate periods of remission and active disease (*Planell et al., 2013*; *Strober, Fuss & Mannon, 2007*). Worldwide, UC is more common

Corresponding author
Changhong Liu, cfzj2019@sdu.edu.cn

than Crohn's disease (CD). Both diseases are more common in industrialized countries, particularly in North America and Western Europe, although their incidence is rising in Asia. The whole morbidity reported is between 1.2 and 20.3 cases per 100,000 persons per year, and the prevalence is between 7.6 and 245 cases per 100,000 persons per year (*Danese & Fiocchi, 2011*; *Loftus, 2004*). No sex preponderance exists in UC (*Bernstein et al., 2006*). The peak age at onset of the disease was 30–40 years (*Cosnes et al., 2011*). A total of 8–14% of patients have a family history of IBD and first-degree relatives to patients with UC have four times the chance of developing the disease (*Childers et al., 2014*). Studies have confirmed that genes, environment, intestinal microorganisms and autoimmune factors are involved in the etiology of UC (*Chu et al., 2016*; *Dignass et al., 2012*). However, the exact pathogenesis of UC is not clear.

With the progress of genome-wide research, more and more genes closely related to UC have been discovered. The research of DNA microarrays by *Lawrance, Fiocchi & Chakravarti (2001)* discovered that the differentially expressed genes (DEGs) in UC inflammatory sites, in addition to the expected variety of cytokine, chemokine related genes, and inflammation-related HNL, NGAL, proliferation-related GRO, as well as the tumor-related DD96, DRAL, MXI1, and immune-related IGHG3, IGLL2, CD74. An RNA Microarray study of IBD, including six UC patients, found that genes related to functions of biosynthetic and metabolic processes, electrolyte transport, such as HNF4G, KLF5, AQP8, ATP2B1, and SLC16A, were significantly down-regulated in UC samples. Nevertheless, the over-expressed genes are mainly involved in such biological processes as Cell motility, Immune and inflammatory response, Antimicrobial response, Regulation of Cell growth and proliferation, and cytokine chemotaxis. For example, CORO1A, MMP12, TIMP1, PTGDS, CD79A, POU2AF1, TNFRSF7, IGFBP5, FSCN1, CCL11, etc (*Wu et al., 2007*). More recently, a similar study involving 67 UC patients showed significantly up-regulation of genes including SAA1, DEFA5&6, MMP3&7, S100A8&9 (*Noble et al., 2008*). A meta-analysis of 2,693 UC patients reported about 30 gene loci closely related to UC, including not only TNFSF15, NKX2-3, IL12B, MST1, IL18RAP, HLA, IBD5, RNF186/OTUD3/PLA2G2E, DLD/LAMB1, IL10, CARD9, IFNG/IL26, JAK2, IL23R, but also novel FCGR2A, 5p15, 2p16, CARD9 and ORMDL3 (*McGovern et al., 2010*). However, genetics only explains 7.5% of the disease variation, with small predictive ability for phenotypes, and are currently limited in clinical practice (*UK IBD Genetics Consortium & The Wellcome Trust Case Control Consortium 2, 2009*).

The aim of this article is to further explore the interaction of genes related to the pathogenesis of UC and the interaction of the enriched signal pathways, elucidating underlying pathogenic events that may contribute to find new and valuable therapeutic targets of the disease.

Gene Expression Omnibus is a public database, and dozens of gene expression datasets about UC patients are freely available, which provide very valuable information, and it could be reused to provide new insights into the molecular pathogenesis of UC. In addition, due to the small sample size in single dataset and discrepancies of the characteristics among multiple heterogeneous datasets, individual genome-wide gene expression datasets could have restricted capability in forecasting the functional gene networks. Thus, it is necessary

to gather those datasets and synthetically integrate those massive data through systems biology tools, and finally receive the stable and credible results (*Marques et al., 2010*; *Rung & Brazma, 2017*; *Seifuddin et al., 2013*).

The robust rank aggregation (RRA) analysis is a strict tool of systems biology, which can be adopted to the comparison of multiple gene ranking lists obtained from experiments on different platforms greatly expanded the sample size, making the identification of genes related to diseases more reliable and valuable (*Kolde et al., 2012*). The theory of RRA is that by looking at the location of genes respectively in each ranked list and comparing it with a randomly shuffled baseline list, each gene will be assigned a *p*-value, and the better the location in these ranked lists, the smaller the *p*-value will be. The final ranking of genes is based on the *P* value, and logarithmic fold changes (logFC) can be calculated as needed to determine the importance of genes together with the *P*-value.

In the current, systematic review and comprehensive integration of genome-wide gene expression datasets in UC is still missing. Therefore, we performed the systematic review and comprehensively integrated those genome-wide gene expression datasets through RRA to identify the most probable causative genes of UC. We hope to mark out some deepening insights into UC pathogenesis and provide some molecular target for therapeutic.

Moreover, we would use weighted gene co-expression network analysis(WGCNA) to categorise those important and aberrantly expressed genes into several biologically functional modules (*Langfelder & Horvath, 2008*; *Prom-On et al., 2010*), which could be biologically meaningful gene clusters and play important roles in UC pathogenesis.

## MATERIALS & METHODS

### Datasets search and eligibility criteria

On the Gene Expression Omnibus (GEO) home page (http://www.ncbi.nlm.nih.gov/geo/), "UC biopsy" was used as the search term, and the datasets in the search results were filtered according to the following criteria: (1) the gene expression profile measured by microarray chip technology; (2) the dataset was a comparison between active UC patients' tissue and non-UC patients' healthy tissue; (3) Sample size should be at least 5; (4) The database provided raw data or gene expression. Fragments Per Kilobase of transcript per Million fragments mapped (FPKM) matrix files for these datasets and can be used for reanalysis. The raw data is the direct information measured by instrument, in CEL format, which can be processed by R and converted into TXT format of gene expression FPKM matrix. The gene expression FPKM matrix files provided by the website should not have been normalized. Datasets that did not meet the above criteria are excluded.

### Robust Rank Aggregation (RRA) analysis

The data set of a single platform is difficult to reach a large sample size, and the result is of low credibility. We used the RRA analysis method to comprehensively compare and analyze the results obtained from the genetic difference analysis of each platform, and selected the genes with strong consistency and difference, so as to make the final differentially expressed genes (DEGs) more convincing. Multiple packages of R software were applied for data processing and statistical analysis (*R Core Team, 2018*; *Gentleman et al., 2004*).

## Affy package for data preprocessing

read.AnnotatedDataFrame(), read in the grouping information file for the samples(UC patients and controls); read.csv(), read in the annotations files of gene expression omnibus platform (GPL), including the conversion of probes to gene symbols; eset.rma <- justRMA(), datExpr=exprs (eset.rma), these two-step functions apply the RMA method to normalize original files, with the purpose of adjusting the overall characteristics of a single sample to make it more suitable for comparison.

## Surrogate variable analysis (SVA) package for batch effect removing

Batch effect is caused by different samples under different conditions such as experiment time, experiment environment, instrument, etc., and merely data normalization cannot remove batch effect. SVA package were used to remove the batch effects from different samples of the same platform (*Chen et al., 2011*; *Leek et al., 2012*). This step is performed using Empirical Bayes method, whose core function is ComBat Finally, gene expression value matrix files with row name as gene symbol and column name as sample number were obtained for each platform for further analysis.

## Limma package for differential genes analysis

The limma package is a comprehensive package with many options for loading data, data pre-processing (background correction, intra-group normalization and inter-group normalization), and differential genetic analysis. The function of empirical Bayes linear regression method for finding differential genes is very popular. At the same time, limma package is very scalable. Both one channel and tow channel data can be analyzed for differential genes, even including quantitative PCR and RNA-seq data types (*Ritchie et al., 2015*).

The gene expression matrix files obtained in the last step were used for differential gene analysis between UC and Control groups by Limma package respectively, so as to acquire the DEGs of each platform (*Wettenhall & Smyth, 2004*). MakeContrasts() as the key function and gene rank lists of different platforms were generated. In the process, the False Discovery Rate (FDR) is calculated by benjamini-hochberg correction method, which means a adjusted P-Value, but the P-Value is still used as the basis for the significance judgment of the result.

## RobustRankAggreg package for RRA analysis

The RobustRankAggreg package was used to implement RRA analysis for Gene rank lists of different platforms to generate the most valuable DEGs (*Kolde et al., 2012*). Core functions: list(), rankMatrix(), aggregateRanks Genes with $P$ value < 0.05 and |logFC|>1 were selected, and the smaller the $P$ value, the higher the ranking, often small $P$ value of the gene corresponds to a large |logFC|. The final result was visualized by pheatmap package.

## WGCNA

In order to clarify the main role of DEGs in the pathogenesis of UC, this method is used to cluster genes with close relationship in the same module. The weighted gene co-expression network was constructed by the WGCNA package of R.

First, an appropriate gene expression FPKM matrix file is required. A number of genes and suitable samples were extracted from the raw data, and the matrix file is the FPKM of these genes for each sample. The DEGs generated in the RRA analysis were only the most important genes, and could not present the overall picture of the co-expression network. In order to cover most valuable difference genes, we adjusted the cut off value to $p < 0.05$ and |logFC|>0.14. In other studies, |logFC| values are often different in order to select sufficient and relatively high value genes for WGCNA. For example, Yan et al. selected |logFC|>0.26 (*Yan et al., 2018*), while *Lu et al. (2014)* set |logFC|>0.585 in order to get more differentiated genes . Besides, only samples from the same platform can be combined for WGCNA. To make the results more convincing, we selected GPL570 with the largest sample size, including 143 UC patients and 79 controls from eight datasets.

Then, hclust() was used to hierarchical clustering of samples by average method and results in the initial sampletree. The following we defined sample clustering height = 80 to remove the isolated samples from the group, so as to obtain a more hierarchical sampletree for further analysis.

The core process of WGCNA is to build a scale-free distributed topological network, making the functional modules developed more cohesive (*Langfelder & Horvath, 2008*). In the view of many relevant references prove that when the scale-free fit index is greater than 0.85, the network already conforms to the scale-free network distribution (*Lancu et al., 2015*; *Zhang & Horvath, 2005*). We set an appropriate soft threshold power value to make the generated Scale free Topology Model Fit >0.85.

Next, module identification was realized by Dynamic Tree Cut method, setting minModuleSize = 30 and deepSplit = 2. Further, mergeCloseModules(), a function that can be merged automatically, completes the merging of similar modules by setting the minimum height for merging modules at 0.3. Finally, some genes that could not be classified into any functional module were uniformly collected into the grey60 module. Incidentally, the colors of each module are randomly assigned.

### Functional enrichment analysis

Functional enrichment analysis was performed by Metascape (http://metascape.org) accord to the genes assigned to each module (*Tripathi et al., 2015*). In the results, the top 10 biological processes with the minimum $p$ value of each module were listed, which reflected the functional characteristics of the modules.

### Statistical analysis

The version of R used for statistical analysis was 3.5.0 (*R Core Team, 2018*). In all cases, $P < 0.05$ was considered statistically significant.

## RESULTS

### UC microarray datasets

In the end, 14 datasets from five platforms were selected. Details of datasets were shown in Table 1, including GSE number, sample size, Source types, detection platform, data file type and authors. In the study, the number of UC patients in each dataset ranged from

**Table 1  Summary of those 14 genome-wide gene expression datasets involving UC patients.**

| | Gene expression Omnibus Series (GSE) number | Samples (UC patients/ controls) | Source types | Gene expression Omnibus Platform (GPL) | Data file type | PMID |
|---|---|---|---|---|---|---|
| 1 | GSE9452 | 8/5 | Colonic biopsies | GPL570 | Raw data (.CEL) | 19177426 |
| 2 | GSE10714 | 3/3 | Colonic biopsies | GPL570 | Raw data (.CEL) | 20087348 |
| 3 | GSE13367 | 16/20 | Colonic biopsies | GPL570 | Raw data (.CEL) | 19834973 |
| 4 | GSE14580 | 24/6 | colonic biopsies | GPL570 | Raw data (.CEL) | 19700435 |
| 5 | GSE22619 | 10/10 | Sigmoid colon | GPL570 | Raw data (.CEL) | 21621540 |
| 6 | GSE36807 | 15/7 | Colon pinch biopsies | GPL570 | Raw data (.CEL) | 24155895 |
| 7 | GSE38713 | 22/13 | Colonic biopsies | GPL570 | Raw data (.CEL) | 23135761 |
| 8 | GSE47908 | 45/15 | Colonic biopsies | GPL570 | Raw data (.CEL) | 25358065 |
| 9 | GSE73661 | 67/12 | Colonic biopsies | GPL6244 | Raw data (.CEL) | 27802155 |
| 10 | GSE59071 | 74/11 | Colonic biopsies | GPL6244 | Raw data (.CEL) | 26313692 |
| 11 | GSE48958 | 7/8 | Colonic biopsies | GPL6244 | Raw data (.CEL) | 25546151 |
| 12 | GSE6731 | 5/4 | Cecum, Sigmoid, Rectum colon | GPL8300 | Raw data (.CEL) | 17262812 |
| 13 | GSE53306 | 16/12 | Colon tissue | GPL14951 | Matrix File (non-normalized.txt) | 26034135 |
| 14 | GSE65114 | 16/12 | Colonic biopsies | GPL16686 | Raw data (.CEL) | NULL |

three to 74, and the number of normal controls ranged from three to 20. The total number of samples enrolled in the final study was 328 UC patients and 138 healthy controls.

## 150 significant Differentially Expressed Genes (DEGs) between UC and non-UC Patients

The top 100 up-regulated genes and the top 50 down-regulated genes by Robust Rank Aggregation (RRA) analysis were shown in Table S1. $P < 6.11E-07$ and $|logFC|>1$ reminded significant differences of the top 100 up-regulated genes. Besides, the top 50 down-regulated genes had significant difference index of $p < 6.32E-07$ and $|logFC|>1$ (Table S1).

In order to highlight the effect of the presentation, Fig. 1 displayed the logFC for unique dataset platforms and multi-dataset platforms of the top 50 up-regulated and top 50 down-regulated genes. Green represents down-regulation and red represents up-regulation. The colors deepen with the increase of $|logFC|$ respectively. The similarity of color saturation reflects the consistency of these important genes in the datasets of each platform.

The expression of the above 100 DEGs in all samples of GPL570 platform was shown in the heatmap (Fig. 2). Among them, the 50 up-regulated genes mainly include: (1) Closely associated with inflammatory response, such as S100A8&9, CXCL1&8&10&11&13, CCL19&20, CHI3L1, IL1B, IL1RN, VNN1, IDO1; (2) MMP1&3&7&9&10&12, PIM2, TIMP1, SERPINB5 are closely related to extracellular matrix organization process; (3) LCN2, SELL, CFB, CD27, CSF3R, C2, LAX1, CFI are associated with immune response; (4) DMBT1, DUOX2& A2, and TNIP3 are associated with viral and other infections; (5) REG1A, REG3A, REG1B, PLAU, TFF1, ADM, WARS are closely related to positive

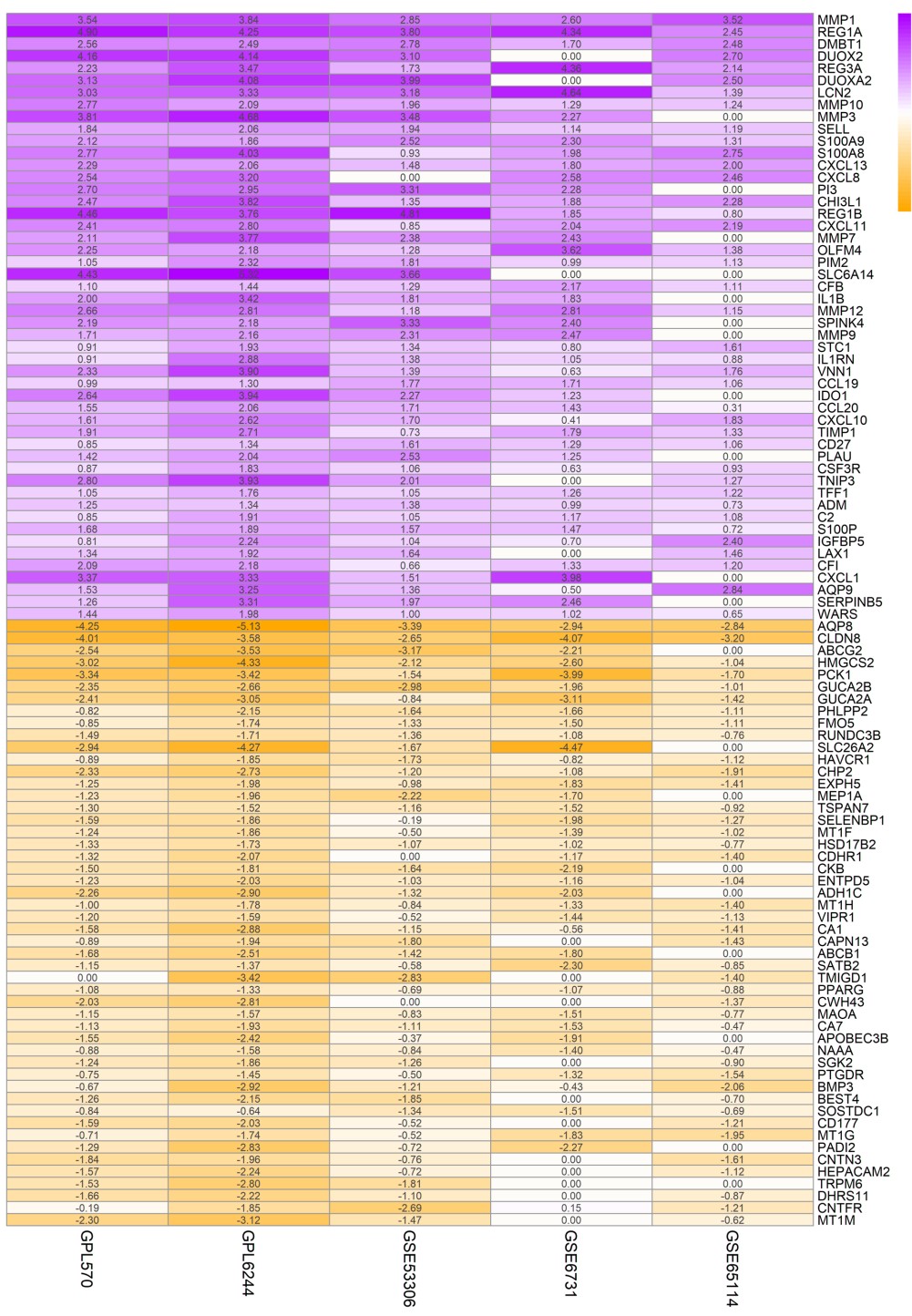

**Figure 1 Top 50 up-regulated and top 50 down-regulated genes in UC.** The vertical axis shows the gene symbols and the horizontal axis represents dataset or merged datasets from same platform. Yellow indicates decreased expression (logFC < 0) and purple indicates increased expression (logFC > 0), the darker the color, the greater the difference; numbers in the figure show the logFC of DEGs, which was calculated by the limma package of R.

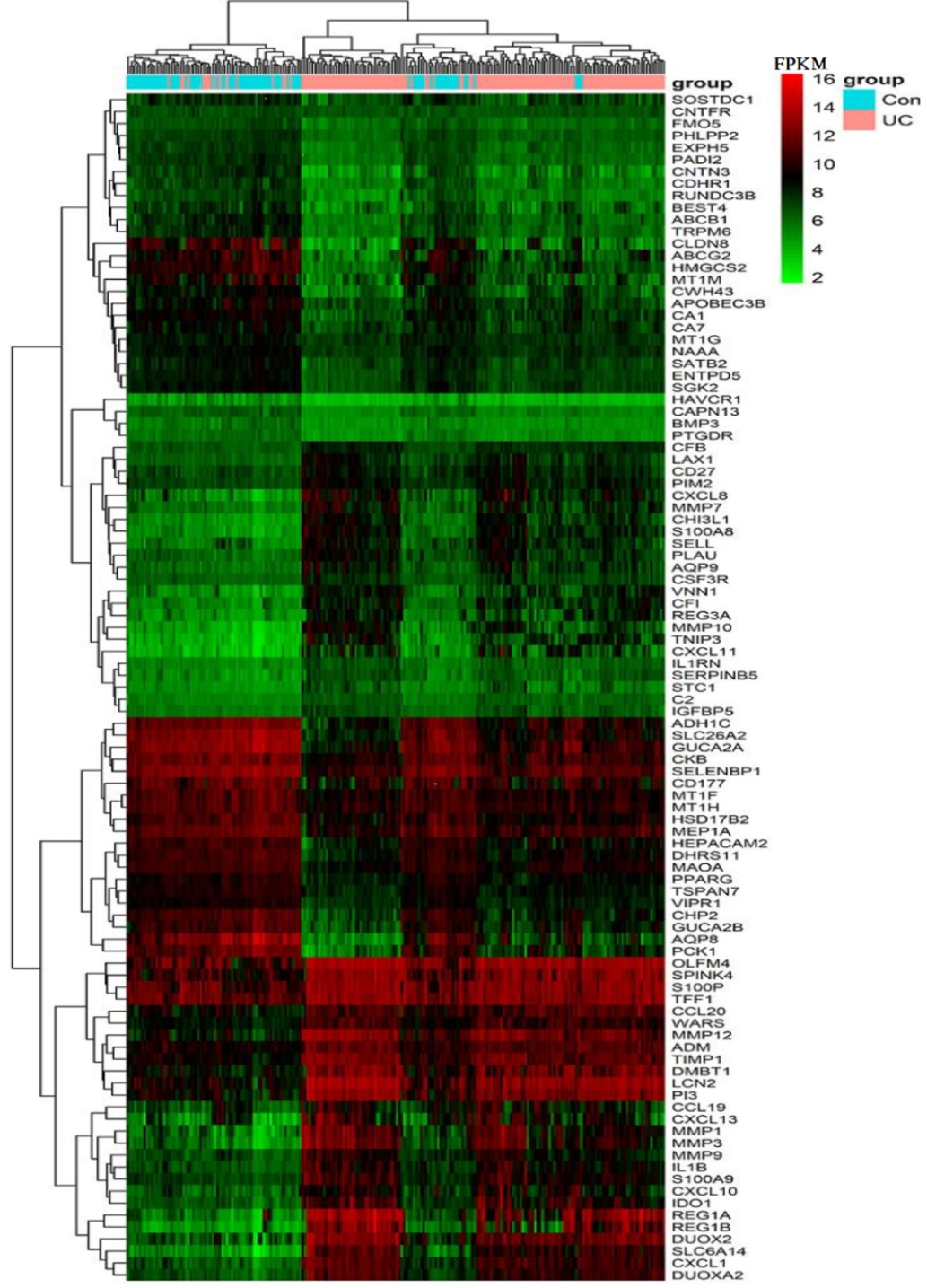

**Figure 2** **Heat map of the FPKM of the top 100 DEGs from GPL570 samples.** The vertical axis lists the gene symbols and the horizontal axis shows the sample grouping, with orange representing the UC group and blue representing the control group. The gradual change in color from green to red in the heat map shows a gradual increase in FPKM of genes. The heat map can roughly distinguish the UC group from the control group.

regulation of cell proliferation process; (6) Other genes with scattered functions, such as PI3, OLFM4, IGFBP5, SPINK4, SLC6A14, and AQP9, are related to cell cycle progression and cell metabolism.

In addition, the other 50 down-regulated genes mainly include: (1) PHLPP2, VIPR1, APOBEC3B, PTGDR were associated with the regulation of immune inflammatory response; (2) AQP8, ABCG2, SLC26A2, CA7, BEST4, TRPM6, which involve in transmembrane transport functions; (3) CHP2, ENTPD5, SGK2, CNTFR, CLDN8, CDHR1, CNTN3, CD177, which are related to cell proliferation, migration, and adhesion; (4) SATB2, PPARG, BMP3, HEPACAM2, GUCA2B, GUCA2A, FMO5, CKB, CA1, MAOA and DHRS11 are closely related to biological processes such as cell cycle and metabolism; (5) Other genes are mainly involved in nutrient metabolism, drug and chemical reactions, and chemotactic regulation of cytokines. And a small number of genes do not yet have a clear function.

Color stratification displayed the difference of expression between the two groups. In the top several genes with the greatest difference, genes such as MMP1, REG1A and AQP8 had been confirmed to abnormally expressed in UC (*Planell et al., 2013*).

## 5,344 DEGs were clustered into seven functional modules through WGCNA

Appropriate samples and genes were screened to construct gene expression FPKM matrix files. Data sets must come from the same platform to be combined into a single matrix file for analysis, and we selected all samples from the GPL570 platform with the largest sample size. After adjusting the cut off value of RRA process to $p < 0.05$ and |logFC|>0.14, 5344 DEGs were obtained, which was more suitable for WGCNA.

When soft-threshold power was set to 10, the scale-free topology index was >0.85, and mean connectivity was infinitely close to 0 (Fig. 3A). The analysis produced 8 co-expression modules, among which seven modules contained more genes and were the main functional modules (Fig. 3B). The number of genes in each module ranged from 97 to 1,718. The module with the largest number of genes was the blue module and the second largest module is the black module with 1,398 genes. Blue and black modules also contain the largest number of the150 most important DEGs. Therefore, we believe that the pathways involved in the two modules dominate the occurrence and progress of UC. The detailed gene names were listed in Table S2. The network heatmap plot showed that these major modules maintain a good independence from each other (Fig. 3C).

## Co-expression modules were enriched to obtain significant pathways

Table 2 listed the functional enrichment analysis results of seven major co-expression modules. Biological processes from were ranked by Log10(P), and having the greatest |Log10(P)| was considered critical.

The genes of the blue module were significantly enriched in the 'extracellular matrix organization, lymphocyte activation, blood vessel morphogenesis, leukocyte migration and inflammatory response'. Also, 'nucleobase-containing small molecule metabolic process, small molecule catabolic process, isoleucine degradation' pathways were the most

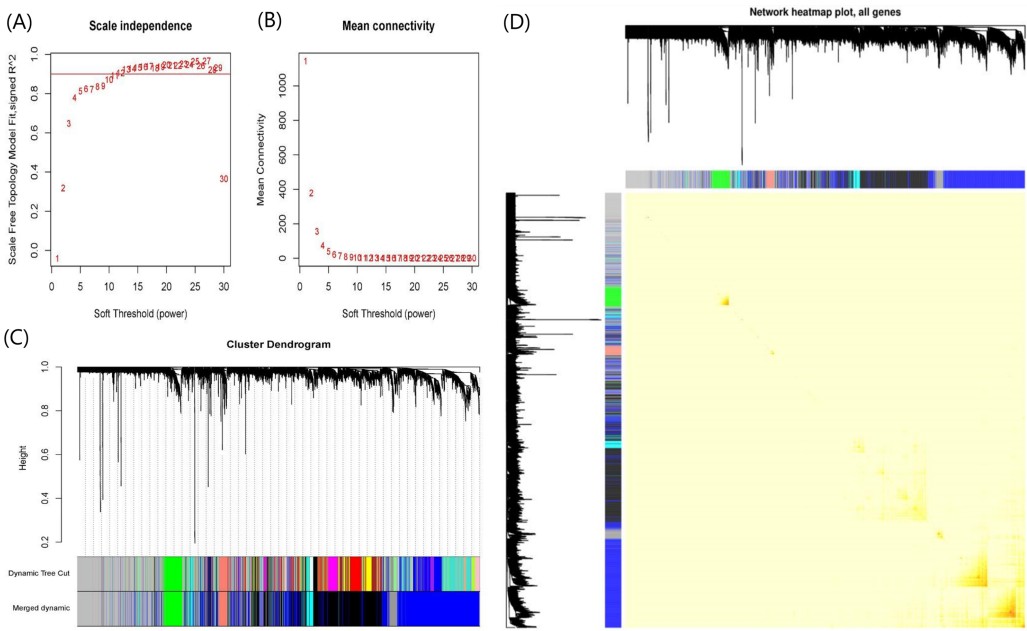

**Figure 3** **Plots in the WGCNA analysis using gene expressions in 328 UC patients and 138 controls from GPL570 datasets.** (A) Influence of soft-threshold power on scale-free topology fit index. (B) Influence of soft-threshold power on the mean connectivity. (C) Cluster dendrogram of coexpression genes and functional modules in UC. More than 15 modules were identified by Dynamic Tree Cutting method with a medium sensitivity (minModuleSize = 30, deepSplit = 2) to branch splitting. Merged Dynamic shows the seven functional modules obtained by merging similar modules in Dynamic Tree Cut (Height > 0.3). (D) The construction of co-expression modules by WGCNA. Each module was assigned a unique color identifier. The progressively saturated red colors indicated the higher overlap among these functional modules.

important pathway enriched in the black module. Besides, the genes of salmon module were mainly enriched in the biological processes of 'interferon signaling, defense response to virus and herpes simplex infection'. The cyan module was enriched into functional pathways involved in multiple fields, including protein regulation, neutrophil immunity, tyrosine kinase pathway, cancer-related pathways and many other aspects. In addition, the enriched pathways of Grey60 and midnightblue modules were closely related to inflammatory response, while the 'Cell Cycle' and 'Cell Cycle Checkpoints' were the results of green module functional enrichment (Table 2).

## DISCUSSION

UC is a type of IBD that affects the large intestine and colon. The pathogenesis of UC is complex and remains largely unknown. It is believed that genetic features, the immune response to microbial dysbiosis, mucosal immune response and environmental factors contribute to the pathogenesis of UC (*Danese & Fiocchi, 2011*). Though many genes have been found be involved in UC, the gene networks associated with the etiology of UC has not been clearly defined.

**Table 2  Pathway and Process Enrichment Analysis of those functional coexpression modules in UC.**

| Modules | GO | Category | Description | Count | % | Log10(P) | Log10(q) |
|---|---|---|---|---|---|---|---|
| Blue module | GO:0030198 | GO Biological Processes | Extracellular matrix organization | 121 | 7.05 | −49.41 | −45.10 |
| | GO:0046649 | GO Biological Processes | Lymphocyte activation | 177 | 10.31 | −48.76 | −44.75 |
| | GO:0048514 | GO Biological Processes | Blood vessel morphogenesis | 166 | 9.67 | −45.01 | −41.40 |
| | GO:0050900 | GO Biological Processes | Leukocyte migration | 123 | 7.16 | −35.59 | −32.42 |
| | GO:0006954 | GO Biological Processes | Inflammatory response | 152 | 8.85 | −29.70 | −26.66 |
| | GO:0001816 | GO Biological Processes | Cytokine production | 143 | 8.33 | −28.99 | −25.98 |
| | GO:0019221 | GO Biological Processes | Cytokine-mediated signaling pathway | 145 | 8.44 | −27.99 | −25.04 |
| | GO:0009611 | GO Biological Processes | Response to wounding | 132 | 7.69 | −25.97 | −23.19 |
| | R-HSA-109582 | Reactome Gene Sets | Hemostasis | 126 | 7.34 | −25.06 | −22.31 |
| | GO:0002250 | GO Biological Processes | Adaptive immune response | 121 | 7.05 | −23.43 | −20.72 |
| Salmon module | R-HSA-913531 | Reactome Gene Sets | Interferon Signaling | 39 | 31.71 | −50.63 | −46.32 |
| | GO:0051607 | GO Biological Processes | Defense response to virus | 30 | 24.39 | −32.91 | −29.64 |
| | hsa05168 | KEGG Pathway | Herpes simplex infection | 21 | 17.07 | −21.50 | −18.36 |
| | GO:0001817 | GO Biological Processes | Regulation of cytokine production | 29 | 23.58 | −18.62 | −15.55 |
| | R-HSA-1280218 | Reactome Gene Sets | Adaptive Immune System | 29 | 23.58 | −16.47 | −13.54 |
| | GO:0060759 | GO Biological Processes | Regulation of response to cytokine stimulus | 16 | 13.01 | −15.40 | −12.52 |
| | GO:0045088 | GO Biological Processes | Regulation of innate immune response | 17 | 13.82 | −11.07 | −8.37 |
| | hsa04621 | KEGG Pathway | NOD-like receptor signaling pathway | 12 | 9.76 | −9.95 | −7.35 |
| | GO:0035455 | GO Biological Processes | Response to interferon-alpha | 6 | 4.88 | −9.15 | −6.62 |
| | GO:0019883 | GO Biological Processes | Antigen processing and presentation of endogenous antigen | 6 | 4.88 | −9.01 | −6.49 |
| | R-HSA-1640170 | Reactome Gene Sets | Cell Cycle | 100 | 35.46 | −82.79 | −78.48 |
| | R-HSA-69620 | Reactome Gene Sets | Cell Cycle Checkpoints | 52 | 18.44 | −44.54 | −40.71 |

| Modules | GO | Category | Description | Count | % | Log10(P) | Log10(q) |
|---|---|---|---|---|---|---|---|
| Green module | GO:0044770 | GO Biological Processes | Cell cycle phase transition | 65 | 23.05 | −44.07 | −40.36 |
| | GO:0051301 | GO Biological Processes | Cell division | 66 | 23.40 | −43.83 | −40.22 |
| | GO:0006281 | GO Biological Processes | DNA repair | 53 | 18.79 | −31.28 | −28.20 |
| | GO:0045787 | GO Biological Processes | Positive regulation of cell cycle | 40 | 14.18 | −24.29 | −21.50 |
| | GO:0051983 | GO Biological Processes | Regulation of chromosome segregation | 24 | 8.51 | −23.76 | −21.00 |
| | GO:0051321 | GO Biological Processes | Meiotic cell cycle | 30 | 10.64 | −20.87 | −18.24 |
| | GO:0045786 | GO Biological Processes | Negative regulation of cell cycle | 43 | 15.25 | −20.64 | −18.02 |
| | GO:0071103 | GO Biological Processes | DNA conformation change | 30 | 10.64 | −19.39 | −16.81 |
| Cyan module | GO:1990778 | GO Biological Processes | Protein localization to cell periphery | 16 | 7.66 | −8.45 | −4.38 |
| | GO:0002446 | GO Biological Processes | Neutrophil mediated immunity | 19 | 9.09 | −6.86 | −3.29 |
| | GO:0030029 | GO Biological Processes | Actin filament-based process | 23 | 11.00 | −6.76 | −3.29 |
| | R-HSA-9006934 | Reactome Gene Sets | Signaling by Receptor Tyrosine Kinases | 18 | 8.61 | −6.76 | −3.29 |
| | hsa04141 | KEGG Pathway | Protein processing in endoplasmic reticulum | 10 | 4.78 | −5.56 | −2.59 |
| | GO:0071407 | GO Biological Processes | Cellular response to organic cyclic compound | 18 | 8.61 | −5.30 | −2.38 |
| | GO:1903829 | GO Biological Processes | Positive regulation of cellular protein localization | 13 | 6.22 | −5.28 | −2.38 |
| | hsa05200 | KEGG Pathway | Pathways in cancer | 14 | 6.70 | −4.83 | −2.02 |
| | GO:0033120 | GO Biological Processes | Positive regulation of RNA splicing | 5 | 2.39 | −4.77 | −1.98 |
| | hsa04810 | KEGG Pathway | Regulation of actin cytoskeleton | 10 | 4.78 | −4.63 | −1.88 |
| | GO:0002274 | GO Biological Processes | Myeloid leukocyte activation | 55 | 35.26 | −45.62 | −41.31 |
| | GO:0006954 | GO Biological Processes | Inflammatory response | 50 | 32.05 | −35.05 | −31.69 |
| | GO:0009617 | GO Biological Processes | Response to bacterium | 40 | 25.64 | −28.23 | −25.09 |
| | GO:0001816 | GO Biological Processes | Cytokine production | 42 | 26.92 | −27.52 | −24.41 |
| | R-HSA-449147 | Reactome Gene Sets | Signaling by Interleukins | 33 | 21.15 | −23.78 | −20.76 |

| Modules | GO | Category | Description | Count | % | Log10(P) | Log10(q) |
|---------|-----|----------|-------------|-------|-----|----------|----------|
| Grey60 module | GO:0097529 | GO Biological Processes | Myeloid leukocyte migration | 24 | 15.38 | −22.95 | −19.96 |
| | hsa04380 | KEGG Pathway | Osteoclast differentiation | 19 | 12.18 | −19.58 | −16.80 |
| | GO:0030099 | GO Biological Processes | Myeloid cell differentiation | 22 | 14.10 | −13.46 | −10.91 |
| | hsa04657 | KEGG Pathway | IL-17 signaling pathway | 13 | 8.33 | −13.26 | −10.73 |
| | R-HSA-6785807 | Reactome Gene Sets | Interleukin-4 and Interleukin-13 signaling | 13 | 8.33 | −12.40 | −9.89 |
| Black module | GO:0055086 | GO Biological Processes | Nucleobase-containing small molecule metabolic process | 128 | 9.23 | −27.65 | −23.33 |
| | GO:0044282 | GO Biological Processes | Small molecule catabolic process | 89 | 6.42 | −24.66 | −21.35 |
| | hsa00280 | KEGG Pathway | Valine, leucine and isoleucine degradation | 25 | 1.80 | −17.79 | −14.86 |
| | GO:0090407 | GO Biological Processes | Organophosphate biosynthetic process | 100 | 7.21 | −17.71 | −14.80 |
| | hsa00071 | KEGG Pathway | Fatty acid degradation | 21 | 1.51 | −14.05 | −11.23 |
| | hsa01200 | KEGG Pathway | Carbon metabolism | 32 | 2.31 | −13.09 | −10.39 |
| | GO:0033865 | GO Biological Processes | Nucleoside bisphosphate metabolic process | 35 | 2.52 | −12.91 | −10.25 |
| | hsa04146 | KEGG Pathway | Peroxisome | 26 | 1.87 | −11.98 | −9.35 |
| | GO:0005975 | GO Biological Processes | Carbohydrate metabolic process | 79 | 5.70 | −11.64 | −9.04 |
| | GO:0008610 | GO Biological Processes | Lipid biosynthetic process | 87 | 6.27 | −10.22 | −7.69 |
| | GO:0006954 | GO Biological Processes | Inflammatory response | 19 | 20.0 | −9.68 | −5.37 |
| | GO:0006959 | GO Biological Processes | Humoral immune response | 13 | 13.68 | −8.73 | −4.78 |
| | GO:0002366 | GO Biological Processes | Leukocyte activation involved in immune response | 17 | 17.89 | −8.52 | −4.78 |
| | GO:0050878 | GO Biological Processes | Regulation of body fluid levels | 12 | 12.63 | −6.04 | −3.05 |
| | GO:0030162 | GO Biological Processes | Regulation of proteolysis | 14 | 14.74 | −5.47 | −2.56 |
| | GO:0045785 | GO Biological Processes | Positive regulation of cell adhesion | 10 | 10.53 | −5.39 | −2.51 |

**Table 2** (*continued*)

| Modules | GO | Category | Description | Count | % | Log10(P) | Log10(q) |
|---------|-----|----------|-------------|-------|-----|----------|----------|
| Midnightblue module | R-HSA-6785807 | Reactome Gene Sets | Interleukin-4 and Interleukin-13 signaling | 6 | 6.32 | −5.27 | −2.42 |
| | GO:0010817 | GO Biological Processes | Regulation of hormone levels | 11 | 11.58 | −5.10 | −2.31 |
| | GO:0045766 | GO Biological Processes | Positive regulation of angiogenesis | 7 | 7.37 | −4.74 | −2.06 |
| | GO:0001666 | GO Biological Processes | Response to hypoxia | 8 | 8.42 | −4.41 | −1.81 |

**Notes.**

'Count' is the number of genes contained in enriched pathway. '%' is the proportion of the total number of genes in each module. 'Log10(P)' is the *p*-value in log base 10. 'Log10(q)' is the multi-test adjusted *p*-value in log base 10.

In this study, 14 genome-wide gene expression datasets were finally included, which involved a total of 328 UC patients and 138 healthy controls. Integrated analysis using the RRA method identified quite a few crucially up-regulated or down-regulated genes (Table 1 & Fig. 1). Some of those genes are novel UC gene signatures and their molecular roles in UC pathogenesis are still largely unknown. These abnormally expressed genes may be therapeutic targets for UC and need further research.

The WGCNA clustering criteria have a great biological significance which have been widely used to explore the molecular mechanisms of various diseases (*Yan et al., 2018*), including IBD (*Lin et al., 2018*; *Xie, Zhang & Qu, 2018*). In our study, the expressions of 5344 UC associated genes obtained from the RRA analysis were used in the WGCNA analysis, together with they were classified into seven co-expression biologically functional modules (Fig. 3B), which highlighted some new insights into the pathogenesis of UC at a systems level.

By functional enrichment analysis of the modules, we revealed several significant pathogenic mechanisms closely related to UC. In the absence of clinical traits, the importance of module is often judged by the number of genes they contain. The blue and black modules both have more than 1,000 genes, and contain the largest number of top 150 genes, which are considered to be the two most important modules.

To further understand the significance of these functional modules in the pathogenesis of UC, enrichment analysis was performed using Metascape. The importance of pathways is based on Log10(P) values. Important pathways in important modules probably have the strongest correlation with the symptoms or pathophysiology of UC. The enrichment analysis of genes in the blue module mainly involved in 'extracellular matrix organization, lymphocyte activation, blood vessel morphogenesis, leukocyte migration' which relevant to inflammatory responses revealed that inflammatory pathway occupies a core position in various pathways related to UC. Extracellular matrix can regulate inflammation, healing and fibrosis. The intestinal extracellular matrix is comprised of various macromolecules, including glycoproteins such as collagens, vitronectin, fibronectin and matricellular proteins. A recent study has reported that extracellular matrix organization strongly promotes the occurrence of Intestinal fibrosis which is common in IBD (*Latella et al., 2014*; *Wynn & Ramalingam, 2012*). The black modules with the second largest number of

genes and the enriched functional pathways mainly include 'nucleobase-containing small molecule metabolic process, small molecule catabolic process, isoleucine degradation'. The regulation of metabolism of various small molecular substances suggests that many pathways and metabolism are active in tissue cells when UC is activated. 'Cell Cycle' and 'Cell Cycle Checkpoints' were the most outstanding pathways of Green module. One study pointed out that the cell cycle regulates the immune, tolerance and autoimmunity functions of T cells, and the excessive inflammation of IBD is the loss of immune tolerance caused by abnormal regulation of the cell cycle (*Sturm et al., 2004*). The enrichment results of Cyan module pathway can be seen that immune response-related pathways are still common and the localization of a large number of proteins inside and outside the cell once again indicates the activity of cell metabolism. In addition, 'pathway in cancer' process conforms to the recognized fact that UC and colorectal cancer (CRC) are closely related. Studies have shown that 8 to 10 years after diagnosis of UC, the risk of CRC begins to increase (*Yashiro, 2014*). Tyrosine kinase receptor pathway, which regulates cell proliferation and differentiation and promotes cell survival, has been closely associated with CRC (*Herr et al., 2018*). Meanwhile, it has been reported that tyrosine kinase receptor RON is highly expressed in UC mucosa (*Hirayama et al., 2007*). Therefore, we believe that tyrosine kinase pathway plays an important role in the occurrence of UC canceration.

Moreover, there are obvious similarities between the pathway enrichment results of grey60 module and midnigntblue module. The former chiefly include 'myeloid leukocyte activation, inflammatory response, response to bacterium'. The latter also focuses on the fields of 'inflammatory response, immune response'. Numerous studies have demonstrated the association between clostridium difficile infection and UC. Clostridium difficile toxins may lead to an enhanced inflammatory response in the presence of Clostridium difficile infection (*Martinelli et al., 2014*). With regard to other bacteria, salmonella and campylobacter infections have also been noted to cause an exacerbation of IBD (*Malik, 2015*; *Singh, Graff & Bernstein, 2009*). The functional enrichment pathways of salmon module mainly involve in 'interferon signaling, defense response to virus and herpes simplex infection', of which 'interferon signaling' is the most important. There were some observational studies on the link between Interferon Signaling and UC. It is generally known that IFN-gamma plays a key role in the early steps of installation of inflammation, promoting monocyte recruitment and activation, and inducing the expression of other inflammatory cytokines. IFN-gamma expression was increased in the pouch mucosa of UC patients compared with controls, and thus it seems to play a pivotal role in UC patients (*Leal et al., 2010*). Interferon signaling has been identified as a central aspect of innate immune response which induces a wide variety of antiviral proteins against pathogens infection. Moreover, interferon signaling play a crucial role in the response to herpes virus infection by antagonizing viral replication and spread (*Noisakran & Carr, 2001*; *Su, Zhan & Zheng, 2016*). This reminds us that the occurrence of UC is probably a sequential process of herpes simplex infection-defense response to virus-interferon signaling in a part of patients. A study reported corticosteroid refractory patients may benefit from antiviral therapy (*Shukla et al., 2015*). This subgroup of patients who were refractory to corticosteroid was likely to undergo above-mentioned sequential process continuously. Therefore, screening
for herpes virus infection, prompt diagnosis and antiviral therapy may effectively relief these patients' condition and reduce colectomy risk. However, the molecular mechanisms underlying the roles of nucleobase-containing small molecule metabolic process in UC are still poorly understood and need to be elucidated in the future.

RRA analysis in the study identifies a large amount of significant DEGs that were drastically up-regulated or down-regulated, plenty of which have been reported in previous articles. We listed top 100 DEGs in the visualization operation to show the reliability of the results. The most significant causative genes are likely to be contained in the top 100 genes and need further experimental verification. Therefore, in our discussion, we will focus on the genes that are considered to be closely associated with the occurrence and development of UC.

MMP1, REG1A and AQP8 have been reported in related literatures (*Planell et al., 2013*). MMP1, which belong to metal dependent enzymes family, is known as interstitial collagenase involved in extracellular matrix turnover (*Fanjul-Fernandez et al., 2009*). MMP1 expression increased in the colonic mucosa of UC patients compared to normal controls, and the mucosa up-regulation of MMP1 correlated with the severity of disease in UC (*Wang, Tan & Zhang, 2009*).There is growing evidence that MMP-1 reflect acute tissue injury and involved in the initial steps of ulceration in UC and new blood vessel formation, but the molecular mechanism underlying its effects remains unclear (*McKaig et al., 2003*; *Wang & Yan, 2006*). In the previous literature it has been pointed out by several authors that Abnormally high expression of REG1A is present in the colonic mucosa in UC patients, but its precise molecular mechanism is far from being completely understood (*Planell et al., 2013*). Currently several researches reported that AQP8 play important roles in gastrointestinal diseases, including UC. The expression of AQP8 is a marker of normal proliferating colonic epithelial cells and AQP8 are closely connected with fluid transport in colon (*Zhao et al., 2016*). A study reported that AQP8 expression reduced in the ileum of UC patients while AQP8 was dramatically induced in the colon of UC patients (*Zahn et al., 2007*). However, a study with larger number of samples found that the AQP8 expression was markedly decreased in UC colon tissue compared to healthy subjects in agreement with the our results (*Min et al., 2013*).The decrease of AQP8 may lead to the disorder of colonic mucosal fluid absorption and reduce the secretion of intestinal tract, but its molecular mechanism is poorly understood (*Calamita et al., 2001*; *Elkjaer et al., 2001*). High expression of DUOX2 and DUOXA2 have been shown in patients with active UC, especially where inflammation is prominent. Both of them are regulated by inflammation and crypt-by-crypt basis in UC tissues, which can increase the production of $H_2O_2$. This process can enhance innate defense, but has the risk of potential DNA damage (*MacFie et al., 2014*). Studies have confirmed that DMBT1 and IL-22 mRNA are obviously highly expressed in UC mucosa, and have a significant correlation. Il-22 increased DMBT1 expression by stimulating STAT3 and NF-$\kappa$B. This process is likely to have an important effect on the innate immunity of UC mucosa (*Fukui et al., 2011*). A study of 32 UC patients found that the detected levels of MMP-9 and LCN-2 in feces of patients with active UC were significantly increased, and that fecal MMP-9 could be a reliable biomarker of IBD activity (*Buisson et al., 2018*). Coincidentally, another report suggested that Serum LCN2
level significantly increased in patients with active UC, and it can serve as a biomarker of active UC (*Stallhofer et al., 2015*).

Among the significantly down-regulated genes, the high ranking ABCG2 also demonstrated low expression in patients with active UC in a previous study. ABCG2 is an efflux transporter involved in mucosal barrier function, low expression of which may increase the risk of tissue exposure to carcinogens, bacterial toxins and drugs (*Englund et al., 2007*). There are also some genes with significant differences, such as HMGCS2 and PCK1 are novel gene signatures of UC, but still lack of direct experimental evidence. Therefore, their relationship and value with UC need to be validated in future studies.

As mentioned above, this study creatively applied the RRA method to comprehensively analyze the DEGs of large samples from multiple platforms. The important DEGs filtered out are more reliable, and the functional distribution of the DEGs is more concentrated, which is conducive to clustering clearer functional modules in the process of WGCNA, so as to reveal an intimate pathway network with UC. The results of our study on important genes are compared with the results of other similar studies in Table S3. It can be found that in some studies, the DEGs of RNA microarray between UC and control group have a great overlap with our result, or at least a similarity in functional distribution.

Since the sample size of our study is larger, the results are more comprehensive. Some unreported genes still have considerable research value due to their homology with many genes that have been confirmed to be closely related to UC in gene function (*Kobayashi et al., 2013*; *Noble et al., 2008*; *Planell et al., 2013*; *Wu et al., 2007*). Compared to other data re-analyses researches on UC, almost all of the studies were conducted directly by functional clustering for a large number of DEGs to display the main mechanisms of the disease, which can not reflect the importance of the individual genes (*Feng et al., 2017*; *Song et al., 2018*). We innovatively used RRA to summarize and analyze the differential genes in multiple data sets to obtain the likely important causative genes of UC.

Regarding this study, findings are consistent with many previous research conclusions and current mainstream views, which reflects the reliability of research methods and results. However, due to various reasons, the research has some limitations. Firstly, because the raw data does not provide enough information about clinical traits and disease outcomes of samples, the correlation degree between modules and clinical traits cannot be analyzed by WGCNA method, which is limited in the judgment of module importance. Secondly, as for the setting of cut off value, $p < 0.05$ is considered to have statistical significance. LogFC is set based on the similar studies and the appropriate number of genes needed for the next analysis. The difference in the value set makes a difference in the results, but not in the essence. Third, the comprehensive analysis of multiple data sets is conducive to the selection of genes with relatively consistent differences for key research. However, there are differences in experimental conditions and sample composition of different data sets, which may cause some valuable information to be cleared in the data processing. Finally, this study only delineates the possible range of closely related genes through bioinformatics method, showing the most important pathways related to the pathogenesis. The results still need to be verified by specific experimental research, and provide help for the progress of disease diagnosis and treatment.

## CONCLUSIONS

Bioinformatics analysis helps us narrow the scope of our research, which deepens the understanding of the molecular mechanism and provides theoretical foundation for molecular target therapy. The biggest characteristic of this study is that in the pathogenesis of UC, immunity and infection are the two most important factors. We suspect that the two are most likely to be cause-and-effect in the process of disease initiation and progression, which is a hot topic in medical research at present. Herpesvirus infection-viral response-interferon pathway may be the trilogy of corticosteroid refractory UC patients, who are necessary to accept antiviral therapy.

We can use this research as the basis for further clinical specimens experiment to verify these genes and pathways, which may lead to future insights into disease pathogenesis, diagnosis, and treatment.

### Funding
The authors received no funding for this work.

### Competing Interests
The authors declare there are no competing interests.

### Author Contributions
- Jie Zhu conceived and designed the experiments, performed the experiments, analyzed the data, contributed reagents/materials/analysis tools, prepared figures and/or tables, authored or reviewed drafts of the paper, approved the final draft.
- Zheng Wang performed the experiments, prepared figures and/or tables, authored or reviewed drafts of the paper, approved the final draft.
- Fengzhe Chen conceived and designed the experiments, prepared figures and/or tables, approved the final draft.
- Changhong Liu conceived and designed the experiments, contributed reagents/materials/analysis tools, authored or reviewed drafts of the paper, approved the final draft.

### Data Availability
Data is available at NCBI GEO: GSE9452, GSE10714, GSE13367, GSE14580, GSE22619, GSE36807, GSE38713, GSE47908, GSE73661, GSE59071, GSE48958, GSE6731, GSE53306, GSE65114.

GPL570 data file after batch effect removing is available at Figshare: Chen, Fengzhe (2019): GPL570 geneMatrix.xls. figshare. Presentation. https://doi.org/10.6084/m9.figshare.8306720.v1

GPL6244 data file after batch effect removing is available at Figshare: Chen, Fengzhe (2019): GPL6244 geneMatrix.xls. figshare. Presentation. https://doi.org/10.6084/m9.figshare.8306843.v1
## Supplemental Information

Supplemental information for this article can be found online at http://dx.doi.org/10.7717/peerj.8061#supplemental-information.

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
