# Peer review of "Identification of genes and functional coexpression modules closely related to ulcerative colitis by gene datasets analysis"

_PeerJ, doi:10.7717/peerj.8061_

## Round 0.1 · original submission · Major Revisions

Both reviewers clearly felt that this manuscript has promise, but also requires substantial revisions to be publishable including grammatical issues, phrasing, presentation, context, etc. Given they seem willing to take another look I am recommending "major revisions" to the PeerJ staff and forwarding their comments on to you. In general, it would help the process if you can include with your revision a response to reviewers as a means to expedite re-review and as a guidepost to address their concerns.

Reviewer 1 ·

Basic reporting

The introduction text should be broken into multiple paragraphs.

The article does not cite previous gene expression studies of ulcerative colitis such as:

* Lawrence, Fiocchi, and Chakravarti. Human Molecular Genetics. 2001.
* Wu, et al. Inflammatory Bowel Diseases. 2007. (This paper is referenced in a table but not cited in the bibliography.)
* Noble, et al. Gut. 2008.

McGovern, et al. Nature Genetics. 2010. discusses a genome-wide association study for UC. Although not gene expression, the part of the introduction mentioning GWAS studies for UC should be expanded, with more details (especially of results) and citations.

The word "biopsy" is misspelled in the methods. The R package name is truncated ("Robust Rank Aggreg" instead of "Robust Rank Aggregation").

Figure 1: Meaning of horizontal axis is not explained.

Results:
Statements such as "Among them, genes such as MMP1, REG1A and AQP8 had been confirmed to abnormally express in UC" should be followed by citations to the original studies.

I think it would be helpful to the reader to explain what the purpose of each method (e.g., RRA, WGCNA) is. It might even be helpful to rename the results subsections to focus on the results rather than the analyses. For example, "150 Genes are Differentially Expressed Between UC and Non-UC Patients", etc.

The language is not always as clear as possible. For example, I had to reread the sentences "By setting p <0.05 and |logFC| >0.14, the number of differential genes result from RRA analysis would increase to 5,344. The expression matrix of these genes in GPL570 samples was performed for WGCNA." a few times to understand that (1) you used less stringent criteria to identify a large set of differentially expressed genes and (2) used the larger set of genes as input for the WGCNA analysis. It might help to avoid phrases like "these genes" and instead by more specific about which genes.

In the methods, you explain why only the GPL570 samples were used for the WGCNA analysis. It would be helpful to restate this in the results.

Experimental design

Your research goal seems to be to identify differentially-expressed genes that are associated with UC and identify their biological function. Your analysis rests on the assumption that integrating multiple data sets will increase accuracy or statistical power versus analysis of a single data set.

The methods say that you used data sets that provided raw data or gene expression matrices. It's not clear to me what the form of the input data was. For example, did you have to compute the log-fold change yourself using something like DESeq? Or, did you use lists of genes with statistically-significant changes in expression as determined by the authors of the original studies? This is particularly pertinent as you are integrating data from multiple studies, which may have been analyzed differently.

Validity of the findings

The manuscript would benefit from more comparison to results from other studies. What do you find that is new compared with the studies done on individual data sets? What do you find is missing from your analysis versus the individual data sets? How much agreement is there between what you find and others find? How much evidence is there that what you find is of biological relevance?

A few times, comments are made on individual genes (e.g., MMP1, REG1A and AQP8) being studied previously but the other studies are not always cited in the same sentence. It would be helpful to compare lists of genes found by the other studies and your studies and compare them. Can you summarize the agreement quantitatively? If possible, put this information in a table.

Some statements do not seem to be supported with evidence or are vague. For example, "blue module was the most important functional module involved in UC." Important is a vague term. What makes it important? Your results seem to suggest that it had the largest number of genes with statistically-significant biological functions. But that does not imply importance.

Additional comments

This is an important study, and I enjoyed reading about your work. I think that the paper would benefit from adding more detail and clearer analysis and comparison to other work.

Reviewer 2 ·

Basic reporting

-. The manuscript explored public genome-wide expression datasets annotated with ‘ulcerative colitis biopsy’ downloaded from ‘Gene Expression Omnibus database’. Authors integrated them to identify genes and pathways differentially enriched between healthy tissue and ulcerative colitis tissue trying to link their finding to pathogenic events that may contribute to new and valuable therapeutic targets.
-. The manuscript has spelling, grammar issues, and unambiguous English used throughout without clear evidence/ground/reference. For example, “in line 106: the absolute value of fold change was used for sequencing”. I don’t understand what this sentence meant. “in line 145: among them, genes such as MMP1, REG1A and AQP8 had been confirmed to abnormally express in UC”. This argument was made without reference. “in line 146: however, the possible value of HMGCS2, PCK1” in the pathogenesis of UC is still the first discovery. Authors pointed out only several genes among many differentially enriched genes. Then, how about others?
-. Authors used symbols without full name of symbols in many places, for example MMP1 and AQP8 in the Abstract.
-. Table 2 should have horizontal lines to distinguish different modules and explanation of table header (for example, count %, log10(p), log10(q)).
-. Figure 1 doesn’t contain legend of heatmap color and explanation of how fold change was calculated for each platform.
-. Figure 2 doesn’t contain legend of heatmap color.
-. Figure 3(B) doesn’t contain explanation of Dynamic Tree Cut and Merged Dynamic.
-. Figure 3(C) doesn’t contain explanation of heatmap color.

Experimental design

-. Authors have clear criteria to generate gene expression datasets comparable for downstream analysis to answer research question.
-. For downstream analysis, authors employed standard tools implemented in R to remove the batched effects from different datasets of the same platform, to identify differentially expressed genes of each platform, to select most valuable differentially expressed genes, and to group important genes into biological functional modules which were annotated by Gene Ontology. However, authors should describe on functions/methods and parameter values being used in detail in each step for reproducibility and comparison with future datasets instead of just mentioning which R packages being used.

Validity of the findings

-. The explanation/rationale of cutoffs on parameters involved in statistical analysis is lacking so that it is difficult to determine how rigorous the results reported would be.
-. Authors provided which public gene expression datasets included for downstream analysis in Table 1. However, authors didn’t provide a final dataset which was quality controlled
considering batch effects and different platforms.
-. Majority of discussion is based on Table 2 which contains functional enrichment analysis. Authors pointed out several pathways as important pathways out of many enriched pathways without the ground (is it based on count/p-value/q-value?) for each functional module.
-. Authors claims that blue module is the most important functional module without the ground.
-. Authors discussed only 4 modules out of 7 functional modules. In conclusion, I don’t see how authors chose modules as important modules, and pathways as important pathways within the modules.

---

## Round 0.2 · Minor Revisions

Both original reviewers have recommended minor revisions. Although you have 40 days, I'd imagine the changes could be done sooner than that

Reviewer 1 ·

Basic reporting

The paragraphs are not well separated. You should either add a blank line between paragraphs or indent paragraphs with a tab.

The English could still use some improvements. For example, the phrase "of which obtained many biological processes through pathway enrichment analysis" from the abstract is not correct. Could this be changed to "which pathway enrichment analysis indicated were associated with many biological processes"?

Re-define terms like DEGs and RRA upon first use in the results and methods. (Don't assume that the reader has read the entire paper.) This will help the reader interpret a given section without needing to refer back to a previous section.

Experimental design

The Limma package supports a large number of different analyses on expression data. More details on the particular analyses and the methods used by Limma are needed.

Would it be appropriate to use a correction for multiple hypothesis testing? (e.g., Bonferroni correction or Benjamini-Hochberg correction). You don't mention this in your Methods section. That said, given the small p-values you are seeing (e.g., 10^-7), I don't suspect that your results will change substantially.

Validity of the findings

The text still has some ambiguities and a few claims that are not well justified. Specifically, what is a "key gene"? I assume that you are looking for causative genes -- genes for which significant up or down regulation leads to the onset of symptoms associated with UC. You should be clearer as to what makes a gene "key."

Secondly, the methods (e.g., rank aggregation of genes based on expression, pathway enrichment analysis) used are based on association testing. Association testing cannot infer causation. Consequently, your analysis may identify genes that are up or down regulated due to side effects (e.g., inflammation) of UC rather than causing the UC symptoms. Be careful about drawing conclusions about causation.

Your analysis produced hypotheses (e.g., relationship between UC and herpes virus) that are interesting and worthwhile for follow up. However, I would be careful about assuming that your analysis conclusively proves anything.

Additional comments

Thank you for your efforts in responding to previous reviewer comments and implementing the request changes. The changes you made a greatly improved the readability and depth of the paper. Your analysis is interesting and a valuable contribution.

Reviewer 2 ·

Basic reporting

-. The manuscript explored public genome-wide expression datasets annotated with ‘ulcerative colitis biopsy’ downloaded from ‘Gene Expression Omnibus database’. Authors integrated them to identify genes and pathways differentially enriched between healthy tissue and ulcerative colitis tissue trying to link their finding to pathogenic events that may contribute to new and valuable therapeutic targets.
-. The author improved the manuscript a lot. I noticed the following sentences which need to be revised: Line 161, Line 168, Line 288, Figure 2 legend, Table 2 legend.
-. Table 1 has a column for Authors. I think that publication ID (for example, PMID) would fit better.
-. Table 2 has legend above and below the table two times. Please remove one.
-. Table 2 has three columns for %, log10(p), log10(q). Please control decimal floating-point numbers.
-. Figure 1 doesn’t contain legend of heatmap color. Please put ‘log10(FC)’ on top of color legend.
-. Figure 2 doesn’t contain legend of heatmap color. Please put legend on top of color legend.

Experimental design

-. Authors have clear criteria to generate gene expression datasets comparable for downstream analysis to answer research question. For downstream analysis, authors employed standard tools implemented in R to remove the batched effects from different datasets of the same platform, to identify differentially expressed genes of each platform, to select most valuable differentially expressed genes, and to group important genes into biological functional modules which were annotated by Gene Ontology.
-. Authors improved the manuscript a lot in explaining methods and parameter values. I have two questions. In line 164, authors said that they used 0.05 of adjusted p-value cutoff and |logFC| > 0.14 because these cutoffs have been used in Yan et al, 2018. The following is the sentence from Yan et al, 2018: “To include enough number of genes into WGCNA analysis, genes with Bonferroni’s adjusted p < 0.05 and logarithmic fold changes (logFCs) >0.26 were selected from the final ranked gene list.” It seems that Yan et al used different cutoffs. Please justify the reason of your cutoffs. In Line 182, authors said that p <0.05 was used for all hypothesis testing. Is this correct? It seems that adjusted q-value was used.
-. In line 169, authors mentioned ‘the ideal size of sampletree’. What do you mean by this?
-. In line 170, authors mentioned ‘scale-free topology index’. Please explain why they tried to generate clusters with certain scale-free topology index.

Validity of the findings

-. Authors diligently validated their findings comparing with results from literatures related to ulcerative colitis.

---

## Round 0.3 · accepted · Accept

I was asked to take over this submission as the previous Editor is unavailable. Thank you for your careful revision and accommodating the reviewers concerns in this second round of review. I feel you've done a fine job of making adjustments to the various comments from these reviewers and your paper is now ready for acceptance.